# *Salmonella* Enteritidis Bacteriophages Isolated from Kenyan Poultry Farms Demonstrate Time-Dependent Stability in Environments Mimicking the Chicken Gastrointestinal Tract

**DOI:** 10.3390/v14081788

**Published:** 2022-08-16

**Authors:** Amos Lucky Mhone, Angela Makumi, Josiah Odaba, Linda Guantai, K. M. Damitha Gunathilake, Stéphanie Loignon, Caroline Wangari Ngugi, Juliah Khayeli Akhwale, Sylvain Moineau, Nicholas Svitek

**Affiliations:** 1International Livestock Research Institute (ILRI), P.O. Box 30709, Nairobi 00100, Kenya; 2Department of Zoology, School of Biological Sciences, Jomo Kenyatta University of Agriculture and Technology (JKUAT), P.O. Box 62000, Nairobi 00200, Kenya; 3Département de Biochimie, de Microbiologie et de Bio-Informatique, Faculté des Sciences et de Génie, Université Laval, Québec, QC G1V 0A6, Canada; 4Groupe de Recherche en Écologie Buccale, Faculté de Médecine Dentaire, Université Laval, Québec, QC G1V 0A6, Canada; 5Félix d’Hérelle Reference Center for Bacterial Viruses, Université Laval, Québec, QC G1V 0A6, Canada

**Keywords:** phages, *Salmonella enterica*, simulated gastric fluid, simulated intestinal fluid, pH stability

## Abstract

Multi-drug resistant (MDR) *Salmonella enterica* Enteritidis is one of the major causes of foodborne illnesses worldwide. This non-typhoidal *Salmonella* (NTS) serovar is mainly transmitted to humans through poultry products. Bacteriophages (phages) offer an alternative to antibiotics for reducing the incidence of MDR NTS in poultry farms. Phages that survive the harsh environment of the chicken gastrointestinal tract (cGIT), which have low pH, high temperatures, and several enzymes, may have a higher therapeutic or prophylactic potential. In this study, we analysed the stability of 10 different *S*. Enteritidis phages isolated from Kenyan poultry farms in different pH-adjusted media, incubation temperatures, as well as simulated gastric and intestinal fluids (SGF and SIF, respectively). Furthermore, their ability to persist in water sources available in Kenya, including river, borehole, rain and tap water, was assessed. All phages were relatively stable for 12 h at pHs ranging from 5 to 9 and at temperatures ranging from 25 °C to 42 °C. At pH 3, a loss in viral titre of up to three logs was observed after 3 h of incubation. In SGF, phages were stable for 20 min, after which they started losing infectivity. Phages were relatively stable in SIF for up to 2 h. The efficacy of phages to control *Salmonella* growth was highly reduced in pH 2- and pH 3-adjusted media and in SGF at pH 2.5, but less affected in SIF at pH 8. River water had the most significant detrimental effect on phages, while the other tested waters had a limited impact on the phages. Our data suggest that these phages may be administered to chickens through drinking water and may survive cGIT to prevent salmonellosis in poultry.

## 1. Introduction

According to a recent FAO report, poultry meat is expected to represent 41% of all the proteins from meat sources in 2030 [1]. In Kenya, poultry farming represents about 30% of the total agriculture contribution to the gross domestic product (GDP), with an estimated 75% of rural families raising chickens [2]. Infectious diseases associated with poultry farming and egg production pose risks to the poultry industry, as well as to farmers and consumers [3]. Non-typhoidal *Salmonella* (NTS) is a common foodborne pathogen transmitted by poultry products [4]. The zoonotic and invasive diseases caused by NTS, such as multi-drug resistant (MDR) *Salmonella enterica* serovar Enteritidis, cause foodborne illness worldwide [5]. It is estimated that *S*. Enteritidis is responsible for over 78 million foodborne diseases globally [6]. It causes high morbidity and mortality, especially among poor, peri-urban populations that lack access to safe water [7,8,9]. In Kenya, invasive MDR NTS accounts for 10.8% and 5.8% of bloodstream infections in children and adults, respectively [10].

At the farm level in Kenya, the current method of preventing or treating salmonellosis in poultry includes using antibiotics, such as fluoroquinolones, or third-generation cephalosporins, such as ciprofloxacin and ceftriaxone [11,12]. These are given as single antibiotics or combined antibiotic therapy [13]. One of the practices in the poultry industry is to provide subtherapeutic levels of antibiotics to poultry in animal feed or water to eliminate pathogens in the gastrointestinal tracts even before the birds show clinical signs, thereby increasing the growth rate of chickens and improving feed conversion efficiency [14]. However, such activities also contribute to the rise in drug-resistant bacterial pathogens [15]. Consequently, the antimicrobial resistance of NTS in poultry is a growing concern because it can negatively affect consumer health when transmitted to humans.

Bacteriophages (phages) offer an alternative to the use of antibiotics to reduce the incidence of MDR NTS [16]. Phages possess properties that make them suitable for *Salmonella* control. They are generally highly specific, self-replicating, self-limiting, and ubiquitous [17,18,19]. The result of bacterial infection by a lytic phage is usually cell lysis [17]. Since their discovery, phages have been used or explored to treat infectious diseases, including NTS [15,18,19,20]. Previous studies have demonstrated that phages given orally to chickens successfully reduce *Salmonella* colonization in the gastrointestinal tract [21,22,23]. It was reported that *Salmonella* phages isolated from abattoirs, chicken farms, and wastewater reduced cell counts of *S*. Typhimurium by over four log_10_ CFU and of *S*. Enteritidis by over two log_10_ CFU within 24 h, when orally administered to chickens in antacid suspensions [24]. It has been shown that using a broad-spectrum phage cocktail is cost effective compared to using a single, narrow-spectrum phage [25]. Encouragingly, the effectiveness of phages in eliminating *Salmonella* sp. Has led to the approval and commercialization of phage-based products [26].

As is the case with pharmaceutical drugs, delivering phages to the exact site of infection remains a hurdle [27]. Phages can be neutralized by gastric hydrochloric acids and digestive enzymes. They can also be rendered ineffective due to fluctuating body temperature during transit in the gastrointestinal tract (GIT) to the small intestine [28]. Simulating GIT conditions may also be used to predict the in vivo behaviours of phage formulations [29]. Biorelevant media, such as simulated gastric fluid (SGF) and simulated intestinal fluid (SIF), which mimic stomach and intestine environments, respectively, have been used to determine the efficacy of phages in vitro and predict their performance in vivo [30,31,32,33].

In this study, we characterized 10 *S*. Enteritidis phages that were isolated from chicken slaughterhouses and poultry farms in Nairobi and Kiambu counties in 2020. The phages were assessed for their ability to persist in different thermal (from 25 °C to 60 °C) and pH (from 1 to 12) conditions, as well as in SGF and SIF. Their persistence in different water sources was also assessed. Finally, the 10 phages were ranked for their ability to persist in the conditions described above using a scoring system to identify those with the highest potential to function in chicken.

## 2. Materials and Methods

### 2.1. Bacterial Strains

*Salmonella* Enteritidis strains used in this study were isolated by collecting 1 g of faecal matter from chickens in farms located in Nairobi and Kiambu counties, which was inoculated in 5 mL of Buffered Peptone Water (BPW) (Oxoid, Hampshire, UK) and incubated overnight at 37 °C. Next, 10 mL of this mixture was added to Selenite Faecal Broth (SFB) (Oxoid, Hampshire, UK) and incubated at 37 °C for 24 h. The samples were then streaked on MacConkey agar media (Oxoid, Hampshire, UK) and subcultured on Brilliance Green *Salmonella* Agar (Oxoid, Hampshire, UK), XLT-4 (Oxoid, Hampshire, UK), and *Salmonella*-*Shigella* Agar (Oxoid, Hampshire, UK). To confirm the identity of *Salmonella*, biochemical identification of the isolates was carried out using Triple Sugar Iron agar (TSI) (Oxoid, Hampshire, UK), Urea hydrolysis test agar (Oxoid, Hampshire, UK), motility indole-lysine media (Oxoid, Hampshire, UK), and BioMérieux API test strips (BioMérieux, Marcy-l’Étoile, France). The isolates were also serotyped using Polyvalent O and H *Salmonella* antisera (*Salmonella* Agglutinating Serum, Remel Europe Ltd, Cambridge, UK.) [34]. All *Salmonella* Enteritidis strains were further confirmed by *invA* PCR [35] and CRISPR typing [36] (Table 1). The Sal568 strain was used to determine phage titres in subsequent experiments, as it is sensitive to the selected phages.

### 2.2. Phage Isolates

Phage samples were first obtained by inoculating faeces, originating from the same chicken farms from which the *Salmonella* strains were isolated, into Tryptic Soy Broth (TSB). After incubation overnight at 42 °C and filtration (0.45 µm Minisart^®^ single-use filter unit, Sigma-Aldrich, Saint Louis, MO, USA), 5 µL of the filtered supernatants were deposited on Tryptic Soy Agar (TSA) plates that contained 5 mL of soft agar (10 mM CaCl_2_, and 0.7% agar) and a 200 µL inoculum of *S*. Enteritidis. The plates were then incubated for 6 h at 42 °C and checked for cell lysis or phage plaques. Phage purification was carried out with five rounds of plaque purification, with a single plaque being randomly selected by round. Phage selection was determined by restriction fragment length polymorphism (RFLP) analysis using the EcoRV enzyme to eliminate very closely related phages. Phage DNA was first extracted with the Phage DNA Isolation Kit (Norgen Biotek Corp., Thorold, ON, Canada) following the manufacturer’s instructions. Then, 20 μL of a reaction mixture that consisted of 1 μg of isolated phage DNA, 1 μL of the restriction enzyme, 2 μL of the Green Buffer (FastDigest) and nuclease-free water was incubated for 2 h at 37 °C. After enzymatic digestion, the phage DNA fragments were separated by electrophoresis in a 0.85% agarose gel in the TAE buffer (40× Tris-acetate-EDTA, Promega, Madison, WI, USA) at 50 V/cm. Biolabs™ 1 kb DNA Ladder was used as a size marker [37].

### 2.3. Phage Stability in pH-Adjusted Media

To determine phage stability at different pH values, the pH of TSB was adjusted by either adding 1 N of sodium hydroxide (NaOH) or 1 N of hydrochloric acid (HCL) until the required pH was obtained (1, 2, 3, 4, 5, 6, 7, 8, 9, and 12). Then, 100 µL of phage lysate at a phage titre of 8.9 × 10^8^ plaque-forming units (PFU)/mL was added to 900 µL of TSB with adjusted pH and incubated at 37 °C for 12 h. Next, serial dilutions were carried out, and PFU/mL were determined using the double-layer technique. To assess the reduction of phage titres during the first 3 h, selected pH values were used (3, 4, 9). Briefly, 100 µL of each phage lysate was added to 900 µL of TSB with adjusted pH and incubated at 37 °C for 3 h. Phage titres were assessed at 0, 0.5, 1, 2, and 3 h, respectively, using the double-layer technique [38,39].

### 2.4. Phage Stability in Simulated Gastric and Intestinal Fluids

Phage stability in simulated gastric fluid (SGF) and simulated intestinal fluid (SIF) was tested as previously described [33,40,41,42]. Briefly, the pH values for SGF (Reagecon co., Shannon, Ireland, DBC12-250) and SIF (Reagecon co., Shannon, Ireland DB13-121) were adjusted to 2.5 and 8, respectively. These are the optimum pH values for the chicken’s true stomach (proventriculus; pH 2.5) and small intestine (pH 8). This was done by adding 1N of NaOH or 1 N of HCL to the solutions. To determine the rate of phage persistence in SGF and SIF, 100 µL of each phage lysate at a titre of 8.9 × 10^8^ PFU/mL was added to 900 µL of the SGF and SIF and incubated at 42 °C for 3 h. Phage titres were checked at 0, 0.5, 1, 2, and 3 h, using the double-layer technique.

### 2.5. Phage Stability in Different Thermal Conditions

The stability of the 10 selected *S*. Enteritidis phages was tested at 25 °C, 30 °C, 37 °C, 42 °C, 50 °C, and 60 °C as previously described [43,44,45,46]. Briefly, 100 µL of each of the *S*. Enteritidis phages at a titre of 8.9 × 10^8^ PFU/mL were incubated overnight at different temperatures. Phage titres were also checked after 0, 0.5, 1, 2, and 3 h. Serial dilution was then carried out, and PFU per ml were determined using the double-layer technique.

### 2.6. Control of Salmonella by Phages in pH-Adjusted Media

To determine the effect of pH on the phages’ capacity to control *Salmonella* sp., the pH of TSB was adjusted by either adding 1 N of NaOH or 1 N of HCL until the required pH was achieved (2, 3, and 8). All phage titres were adjusted to 4.5 × 10^7^ PFU/mL. A culture of *Salmonella* strain Sal568 was grown exponentially for 2 h at 37 °C until 10^6^ colony-forming units (CFU)/mL was reached. Then, 10 µL of the phage lysates were added to 1 mL of the bacterial culture and incubated for 15 min at 37 °C. The mixture was centrifuged at 7000× *g* for 2 min, and the phage-infected cell pellet was resuspended in 1 mL of pH-adjusted TSB. Optical density (OD_600 nm_) was then read at 0, 0.5, 1, 2, 3, and 4 h, as described elsewhere [24,47].

### 2.7. Control of Salmonella by Phages in SGF and SIF

To determine the effect of SGF and SIF on phages’ capacity to control *Salmonella* sp., the pH of SGF and SIF were adjusted to pH 2.5 and pH 8, respectively, by either adding 1N of NaOH or 1 N of HCL until the required pH was achieved. All phage titres were adjusted to 4.5 × 10^7^ PFU/mL. Briefly, a culture of the *Salmonella* Enteritidis strain Sal568 was grown exponentially for 2 h at 42 °C until 10^6^ CFU/mL was reached. Then, 10 µL of the phage lysates were added to 1 mL of the bacterial culture before being incubated for 15 min at 42 °C. The mixture was then centrifuged at 7000× *g* for 2 min, and the phage-infected cell pellet was resuspended in 1 mL of SGF or SIF. Optical density (OD_600 nm_) was then read at 0, 0.25, 0.5, 0.75, 1, 2, 3, and 4 h, as described by others [24,47].

### 2.8. Phage Replication in SGF

To determine the effect of SGF on phage titres following replication, a previously described protocol was used with minor modifications [23]. Briefly, pH-adjusted SGF (pH 2.5) was used, and initial phage titres were adjusted to 2.1 × 10^7^ PFU/mL. *Salmonella* Enteritidis strain Sal568 was grown exponentially for 2 h at 42 °C until 10^6^ CFU/mL was reached. Then, 10 µL of the phage lysate were added to 1 mL of the bacterial culture before being incubated for 15 min at 42 °C. The mixture was then centrifuged at 7000× *g* for 2 min, and the cell pellets were resuspended in 1 mL of SGF. The mixture was incubated at 42 °C while shaking at 200 rpm. Every 15 min, the mixtures were centrifuged at 7000× *g* for 2 min to concentrate the phage-infected cells while collecting 20 µL of the supernatant to check for phage titres on TSA plates by the double-layer technique. The volume of the mixture was maintained by adding 20 µL of SGF. This procedure was repeated after 30, 45, and 60 min post-incubation.

### 2.9. Phage Persistence in Different Water Sources

Water samples were obtained from a river flowing on the ILRI campus (1.2706° S, 36.7240° E), rain from Kang’undo, Nairobi (1.3056° S, 37.3453° E), a borehole from the ILRI farm and the tap from the ILRI laboratories. Upon collection, the waters were divided into three groups: raw, filtered, and autoclaved. After water treatments, 100 µL of each phage (adjusted to 4.5 × 10^10^ PFU/mL) were added to 900 µL of water and incubated at 37 °C. Phage spot assays were carried out after 12, 24, and 48 h, during which 20 µL of the content was collected, serially diluted, and spotted on TSA plates using *S*. Enteritidis Sal568 as a host. PFUs per ml were determined using the double-layer technique [48,49,50].

### 2.10. Data Analysis

A two-way analysis of variance (ANOVA) was carried out to determine the differences in means among phages and time points, as well as upon exposure to different pH and temperature values. A simple linear regression model was used to determine phage replication in pH-adjusted media, SGF, and SIF, and to measure phage persistence in various water sources. Statistical analyses were performed using the GraphPad Prism software version 9.2.0. A *p* value of less than or equal to 0.05 was considered significant for each statistical analysis performed. Experiments involving phages were repeated twice with triplicate values.

## 3. Results

### 3.1. Isolating and Characterizing the Biological Materials

To identify unique *S*. Enteritidis-specific phages from our collection of isolates from Kenyan poultry farms, we first determined the tropism of 63 purified isolates (labelled ILRI_K1 to ILRI_K63, indicating the place of isolation [ILRI] and the country of origin [Kenya]; only the latter was kept in the figures for ease of visualization.) to identify those that have specificity toward *S*. Enteritidis strains. Interestingly, we isolated *Salmonella* phages from about two-thirds of the visited farms while we could recover *Salmonella* sp. strains from only 10% of them. These purified phages were screened against a panel of 16 *Salmonella* strains that were isolated from the same poultry farms, and which belong to the Enteritidis, Heidelberg, and Kentucky serovars (Table 1 and Figure 1A). Using host range and RFLP analyses, 10 unique phages were selected for further characterization (Figure 1B and Table 2). Preliminary whole genome sequencing data analyses further indicate that these are novel phages not hitherto isolated or reported (Table 3).

### 3.2. Assessing Phage Stability in Media at Low and High pH

To identify phages that can persist in the harsh environments of the chicken gastrointestinal tract (cGIT), we first assessed the stability of these 10 phages in low pH conditions found in cGITs. After 12 h of incubation in pH-adjusted TSB, we observed that all phages were relatively stable between pH 5 and 9, with maximum stability around neutral pH (Figure 2A). Most phages were inactivated after 12 h at pH 1 and 2 (Figure 2A). In fact, a total inactivation was observed after only 30 min at pH 1 and 60 min at pH 2 (data not shown). At pH 3, phage titres were significantly decreased after 12 h (Figure 2A). We then looked at individual phage data for the specific pH values of 3 and 9, which are close to values found in the chicken proventriculus (pH between 2 and 3) and intestine (pH between 8 and 9). We observed that all phages behaved similarly with inactivation over time (Figure 2B,C and Appendix A). When comparing among phages, phages ILRI_K11 and ILRI_K14 were inactivated slightly more rapidly after 2 h at pH 3 (Figure 2B). At pH 9, phage titres decreased for up to 3 h (Figure 2C). However, viral titres were significantly higher than those measured at pH 3 (Figure 2C). At pH 9, there were no significant differences among phages within each time point (Appendix A).

### 3.3. Assessing Phage Stability in Simulated Gastric and Intestinal Fluids

We also evaluated the phages’ capacity to remain infectious in commercial simulated gastric (SGF) and intestinal (SIF) fluids. For SGF, the product contained hydrochloric acid, sodium chloride, pepsin, and distilled water. For SIF, the same ingredients were used in addition to potassium phosphate monobasic, sodium hydroxide, and pancreatin. The mixtures aimed to mimic the conditions found in the cGIT.

As indicated above, the chicken proventriculus (true stomach) has a pH between 2 and 3. Therefore, we subjected the 10 phages to SGF conditions at pH 2.5 for 60 min, which is the average transit time of food within this organ [51]. We also performed the experiment at 42 °C which is the average temperature of chickens. Following a 60 min incubation period, a significant drop in phage titre of approximately five logs occurred before stabilizing (Figure 3A). In fact, a decline of approximately three logs occurred after the first 2 min. Phage ILRI_K22 was the most unstable in SGF (pH 2.5), with a final titre of 1 × 10^3^ PFU/mL after 60 min of incubation (Figure 3A). More extended incubation periods completely neutralized the phages (data not shown). There were significant differences among phages during the first 40 min (*p* values ranging from <0.001 to 0.0475, Appendix A). Still, the differences were not significant in the last 20 min (*p* values ranging from 0.0545 to >0.9999, Appendix A).

The potential hosts for these phages can be found in the chicken cecum (intestine), which is a more basic environment with a pH of about 8. We, therefore, also subjected the 10 phages to SIF adjusted to pH 8 for 120 min, which is the average transit time of food in this organ. A drop in phage titre was observed as early as 30 min into the incubation period (Figure 3B). Nevertheless, all 10 phages were relatively stable in SIF for up to 2 h. Phage ILRI_K22 had the lowest phage titre at 120 min (3.1 × 10^6^ PFU/mL), whereas phage ILRI_K6 and ILRI_K47 had the highest final titre at 9.3 × 10^6^ PFU/mL. Phage concentrations showed variable significant differences among phages, with time points at 30, 60, and 90 min of incubation showing the most remarkable significant differences (*p* values ranging from <0.0001 to >0.0472, Appendix A).

### 3.4. Phage Stability at Different Temperatures

Another critical parameter to examine is the ability of phages to remain stable over a range of temperatures. The range included 25 °C, which is the average ambient daily temperature in a large part of Kenya during a significant part of the year [52], 42 °C, which is the average body temperature of chickens [53], and 50–60 °C, which are temperatures that can be reached during phage production processes, such as spray drying [54]. Overall, the phages were relatively stable between 25 °C and 37 °C after 12 h (Figure 4A). At 37 °C, a one log drop in phage titre was observed after 3 h (Figure 4B). There was a significant difference among phages for up to 1 h of incubation (*p* values ranging from <0.0001 to 0.0493, Appendix A), after which there were no significant differences observed among phages (*p* values ranging from 0.0582 to >0.9999, Appendix A). At 42 °C following 3 h of incubation, phage titres were relatively similar, as phage ILRI_K6 had the lowest concentration at 7.5 × 10^7^ PFU/mL, while phage ILRI_K47 had the highest concentration at 8 × 10^7^ PFU/mL (Figure 4C).

While phages remained relatively stable at 37 °C and 42 °C, we saw a significant drop in phage concentration as quickly as one hour into the incubation period at 50 °C (Figure 4D). Phages ILRI_K1, _K3, and _K11 had the lowest concentration (Figure 4D). However, among all of the phages, ILRI _K26 and _K47 were still present at relatively high concentrations after 3 h at 50 °C. At this temperature, significant differences were observed among phages only between 0 and 30 min of incubation (*p* values ranging from <0.0001 to 0.0243, Appendix A). After that point, no significant differences were observed among phages (*p* values ranging from 0.0518 to >0.9999, Appendix A).

### 3.5. Control of Salmonella by Phages in pH-Adjusted Media

Phages can encounter their target bacteria in an animal host and replicate, thereby reducing the targeted bacterial population. We, therefore, tested how a bacterial host might be influenced in low and high pH-adjusted TSB media in the presence of phages. We used *S*. Enteritidis isolate 568 (Sal568) as it is sensitive to the selected phages. Bacterial growth at 37 °C was measured by optical density (OD_600 nm_) in TSB at pH 2, 3, and 8 and in the presence of each phage. In TSB at pH 2, the OD remained constant for up to 4 h of incubation (Figure 5A). There were statistically significant differences observed among the majority of phages at pH 2 from 30 min to 4 h (*p* values ranging from <0.0001 to 0.0448, Appendix A). At pH 3, the OD_600 nm_ gradually increased for one hour, then remained stable for up to 4 h of incubation (Figure 5B). There were statistically significant differences observed among most phages at pH 3 from 30 min to 4 h of incubation (*p* values ranging from <0.0001 to 0.0497, Appendix A). At pH 8, the OD_600 nm_ of Sal568 in the presence of phages dramatically decreased in less than one hour before gradually increasing from two to four hours of incubation (Figure 5C). There were statistically significant differences observed among the majority of phages at pH 8 from 30 min to 4 hours (*p* values ranging from <0.0001 to 0.0494, Appendix A). At pH 2, the phage that best controlled Sal568 growth at the end of the incubation was ILRI_K1; it was ILRI_K9 at pH 3 and ILRI_K11 at pH 8.

### 3.6. Control of Salmonella by Phages in SGF and SIF

We then performed the same experiments described above in SGF and SIF. However, the incubation temperature was 42 °C, and the incubation time was 60 min and 3 h in SGF and SIF, respectively, to mimic the conditions and transit time in the chicken organs represented by these biorelevant dissolution media. The OD_600 nm_ of Sal568 (10^6^ CFU/mL) and the 10 phages (4.5 × 10^7^ PFU/mL) in SGF adjusted to a pH of 2.5 remained constant for up to 1 h of incubation (Figure 6A). Phage ILRI_K22 was the best phage in controlling the growth of the selected *Salmonella* strain in this environmental condition, whereas phages ILRI_K24 and _K26 were the least efficient.

The growth of Sal568 in the presence of each of the 10 phages in SIF adjusted to pH 8 significantly decreased in less than an hour and remained low for up to 3 h of incubation (Figure 6B). Phage ILRI_K11 was less efficient in controlling *Salmonella* growth in this environmental condition compared to the other nine phages, and ILRI_K1 was the most efficient (significances of differences are presented in Appendix A).

### 3.7. Phage Replication in SGF

Since all 10 phages were affected mainly by the conditions encountered in SGF at pH 2.5 (Figure 3A), we tested how phage titres might be influenced in SGF adjusted to pH 2.5 and in the presence of their bacterial host. We followed the same experimental protocol as for the experiments to control *Salmonella* growth in SGF but measured viral titres instead of the optical density due to bacterial growth. We observed that the viral titres were reduced by 0.5 log PFU/mL in the first 15 min of replication. After those 15 min, the viral titres remained constant for 45 min before gradually increasing (Figure 7). There was a significant difference between phage ILRI_K1 and ILRI_K11 (*p* = 0.042) and between ILRI_K9 and ILRI_K11(*p* = 0.0471) at 15 min. Significant differences were also observed between ILRI_K11 and ILRI_K26 (*p* = 0.0356), as well as ILRI_K14 and ILRI_K26 (*p* = 0.0356) at 30 min of incubation. In addition, phage ILRI_K47 had the highest titre and ILRI_K9 the lowest titre after 60 min of incubation.

### 3.8. Phage Persistence in Water from Different Sources

Previous research has demonstrated the feasibility and ease of delivering phages through the water given to chickens in poultry farms [55]. Since the water source for chickens can vary from one farm to another in Kenya, we tested the persistence of a subset of these phages (ILRI_K1, ILRI_K6, ILRI_K14, ILRI_K24, and ILRI_K47) in different water sources, including in rivers, rain, boreholes, and tap water (Figure 8). Water samples from all four sources were tested raw (unmodified samples), filtered, or autoclaved. River water had the most considerable negative effect on phages, with an average reduction of five logs PFU/mL after 50 h of incubation (Figure 8A). Rain (Figure 8B), borehole (Figure 8C), and tap water (Figure 8D) only showed a two log PFU/mL reduction. Autoclaved or filtered river water still significantly reduced phage titres by six and five logs, respectively (Figure 8A).

On average, phage ILRI_K47 had the highest phage titre in all water sources after 12 h of incubation (river: 3.8 × 10^5^ PFU/mL, borehole: 3.2 × 10^8^ PFU/mL, rain: 3.4 × 10^8^ PFU/mL, tap: 5.9 × 10^8^ PFU/mL). On the other hand, phage ILRI_K14 had the lowest average phage concentration after 12 h of incubation in all water sources (river: 1.2 × 10^4^ PFU/mL, borehole: 1.1 × 10^8^ PFU/mL, rain: 1.2 × 10^8^ PFU/mL, tap: 1.2 × 10^8^ PFU/mL).

## 4. Discussion

With the emergence of antimicrobial-resistant bacteria, the use of phages as antibacterial agents is being revisited. To target bacteria in the gastrointestinal tract (GIT) of animals, identifying phages able to reach and persist in those harsh environments is crucial for developing successful and targeted bactericidal interventions. This study aimed to determine if a group of 10 phages isolated from slaughterhouses and poultry farms in Kenya would be amenable for poultry applications.

We observed that all 10 phages were more stable at temperatures ranging from 25 to 42 °C and at pH values ranging from 5 to 9. Since the GIT is more complex than culture media, commercial simulated gastric fluid (SGF) and simulated intestinal fluid (SIF) were also tested. These fluids mimic the environments of the stomach (SGF, pH 1.5–3.5) and intestines (SIF, pH 6.5–8.5). We observed that phages persist for only 20 min in SGF, after which they rapidly lose infectivity. However, phages were stable for a more extended period in SIF. This is consistent with previous findings. For instance, *Vibrio vulnificus* phage titres were reduced by three logs within 2 min in SGF at pH 2.5 [41]. *Salmonella* phage Felix O1 lost infectivity after 10 min of exposure at pH 2, while at pH 2.5, it lost infectivity within 1 h of exposure [56]. Titres of coliphages ΦJLA23, ΦKP26, ΦC119, and ΦE142, exposed to SGF at pH 2.5, persisted after 2, 5, and 15 min, but dropped to undetectable levels after 30 min. On the other hand, in SIF, these coliphages remained stable for 3 h before dropping by two logs [42].

The physiochemical conditions of the GIT naturally aid digestion but have potentially adverse effects on phages [57]. *Salmonella* phages en route to the small intestine, the site of *Salmonella* infection, face various conditions that reduce phage infectivity [58]. In the stomach, the gastric pits release hydrochloric acid and several enzymes, which can denature the phage structural proteins and inactivate the virions [19,33,58,59,60,61]. By changing the protonation state of charged residues, the low pH can affect the complex nature of phage protein interactions. As the charge distribution changes, modifications occur to both the strength and geometry of electrostatic interactions that are essential to protein interactions at low salt concentrations [58].

The effect of pH on phage replication was also tested. We observed that acidic media (pH 2 and 3) affects the phage efficacy in controlling *Salmonella* growth (Figure 3, Figure 4 and Figure 5). In contrast, an alkaline environment (pH 8) did not affect the phage replication process (Figure 5) as much as what was also reported by others [62]. A similar outcome was also observed in SGF and SIF, whereby the SGF reduced phage efficacy while SIF did not (Figure 6). The ability of a phage to persist in an acidic environment is one of the key characteristics used for phage selection [58], as they are more likely to persist in the harsh gastrointestinal environment, which contains hydrochloric acid, enzymes, and bile salts [63].

The body temperature of animals is another crucial parameter that affects phage–bacteria interactions as it plays a fundamental role in phage adsorption, replication, burst size, and length of the latent period [64,65]. Temperatures outside the optimal growth temperature of the bacterial host often result in slower viral replication cycles [61,66,67]. All phages in this study demonstrated high titres at temperatures ranging from 25 °C to 42 °C. However, phages started losing infectivity at 50 °C. This is consistent with previous studies on *Salmonella* phages, which demonstrated that temperatures above 50 °C yielded low phage titres [48,49,50]. However, some phages are known to persist at higher temperatures. An example is the narrow-spectrum phage LSE7621, which effectively lysed *Salmonella* Enteritidis and showed good thermal stability at temperatures ranging up to 50 °C [43]. Higher temperatures (mostly above 60 °C) can denature proteins, resulting in loss of viral infectivity [68,69,70].

Water is often considered the preferred vehicle to deliver phages targeting poultry gastrointestinal pathogens [55,71]. We tested the effect on phage stability of different water sources that might be used in poultry farming (river, rain, borehole, and tap water). As observed in our study and by others [49], river water has a more detrimental effect on viruses compared to groundwater and tap water. This is likely because river water has many organic compounds. River water also undergoes constant fluctuations in pH and temperature, which may directly impact the phage structure [57,63,72]. Boiling river water breaks down the complex organic compounds and leaves behind ions that make water more acidic, resulting in conditions that can have a more detrimental effect on phage infectivity compared to raw and filtered water [50,63,73]. It should be noted that cations, such as calcium and magnesium ions, may also promote phage adsorption to the host bacteria, facilitating the multiplication of viruses [66,67,68,74]. Taken altogether, the persistence of phages in water sources depends on several factors, including their association with solids, the presence of organic matter, ultraviolet light, temperature, and pH, as well as the concentration and type of ions [49,57,69,70].

To summarize our results, we developed a scoring system (Figure 9A) using all parameters, except survival in water sources, to identify the best phages that are likely to function better in vivo. The characteristics used were stability in different temperatures (37 °C, 42 °C, and 50 °C), in TSB media (pH 3 and 9), in SGF (pH 2.5), and in SIF (pH 8), as well as the phages’ capacity to control *Salmonella* and replicate in these media. The phages were ranked under each of those conditions, from highest (score of 1) to lowest (score of 10), and all the scores were added. The phage with the lowest total score was ranked number 1, while the one with the highest total score was ranked last, at number 10. Phage ILRI_K47 was the most robust among all 10 phages tested. In contrast, phage ILRI_K14 had the lowest final titres for most parameters (Figure 9B).

In summary, we have characterized 10 *Salmonella* Enteritidis phages selected from a cohort of 63 phages based on RFLP patterns and demonstrated that they were stable in pH values ranging from 5 to 9 and temperatures ranging from 25 °C to 42 °C. This study also showed that the phages rapidly lost infectivity in SGF but were relatively stable in SIF. We observed that their replication was significantly reduced in media with low pH, as well as in SGF, while their replication was less affected in media with high pH and in SIF. Moreover, we determined that river water had the highest negative effect on phage titres. A scoring system was designed to rank phages for their capacity to survive the overall harsh environment encountered in the chicken GIT. These results suggest that a number of these phages could have a chance of surviving in vivo in chickens.

## Figures and Tables

**Figure 1 viruses-14-01788-f001:**
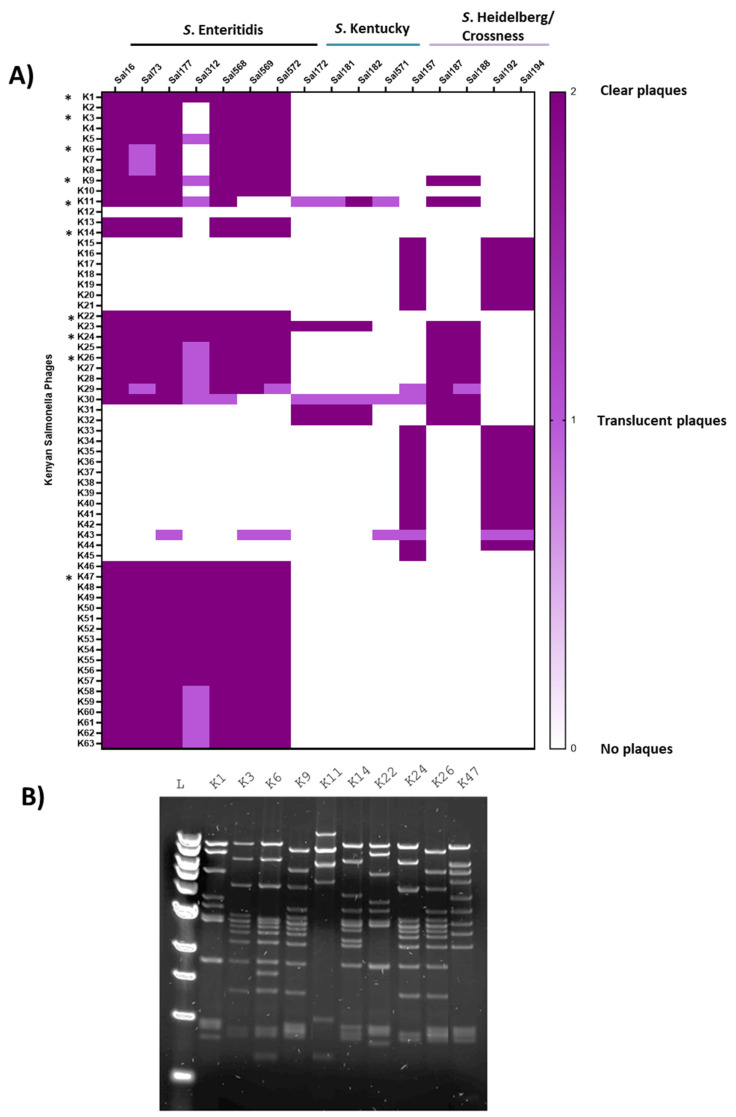
*Salmonella* phage host range and restriction fragment length polymorphism (RFLP) of their genomic DNA. (**A**) A heatmap showing a set of 63 isolated *Salmonella* sp. phages and their tropism for the Enteritidis, Kentucky, and Heidelberg serovars. Dark purple colour indicates clear plaques, violet colour indicates translucent plaques, and white stands for no plaques. The asterisk (*****) indicates the selected phages. (**B**) Gel electrophoresis of the DNA digested with EcoRV from the genomes of Kenyan *S*. Enteritidis phages revealing 10 DNA profiles.

**Figure 2 viruses-14-01788-f002:**
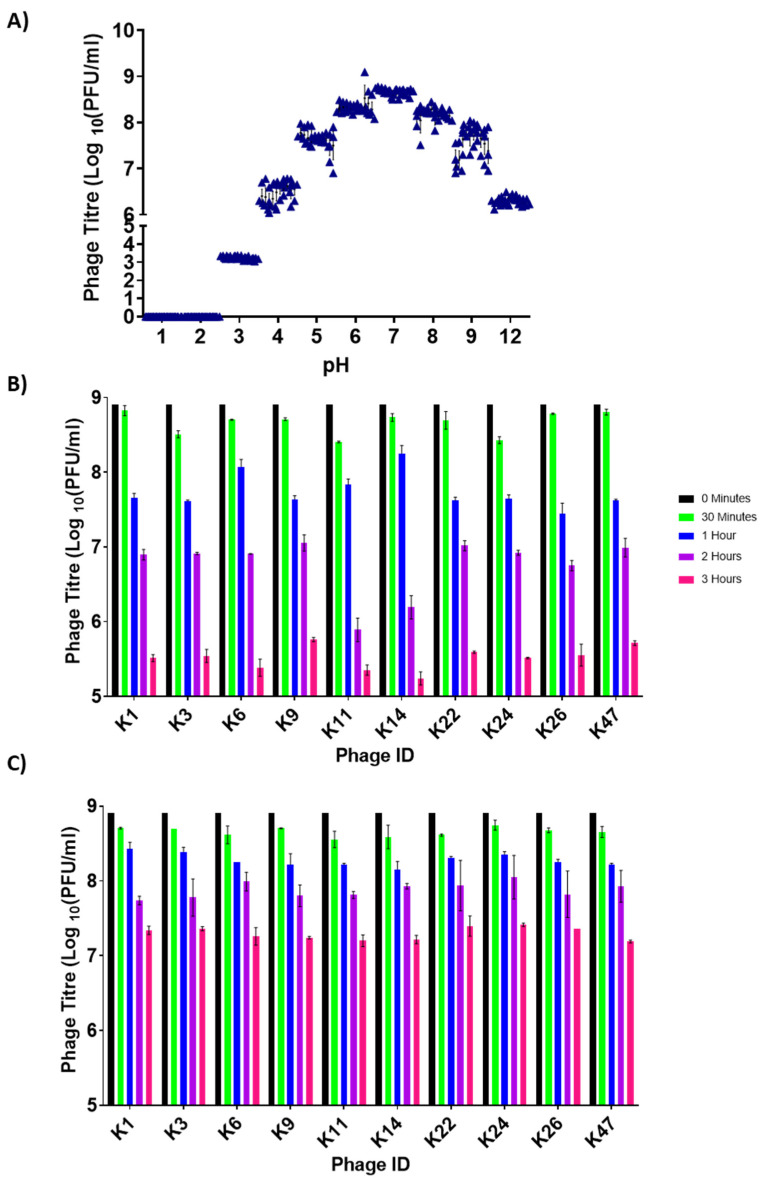
Phage stability in pH-adjusted TSB medium. (**A**) Cumulative stability of *S*. Enteritidis phages in TSB adjusted to pH values of 1, 2, 3, 4, 5, 6, 7, 8, 9, and 12 after 12 h of incubation at 37 °C. The black triangle indicates individual replicate values of phages on each bar graph. (**B**) Stability of individual *S*. Enteritidis phages in TSB at pH 3 for up to 3 h of incubation at 37 °C. Each bar indicates phage titres at specific times. (**C**) Stability of individual *S*. Enteritidis phages in TSB at pH 9 for up to 3 h of incubation at 37 °C. Each bar indicates phage titres at specific times. Error bars represent the standard error of the mean (±SE). Black bar: 0 min, green bar: 30 min, blue bar: 1 h, purple bar: 2 h, magenta bar: 3 h. All experiments were repeated twice and measured in triplicate.

**Figure 3 viruses-14-01788-f003:**
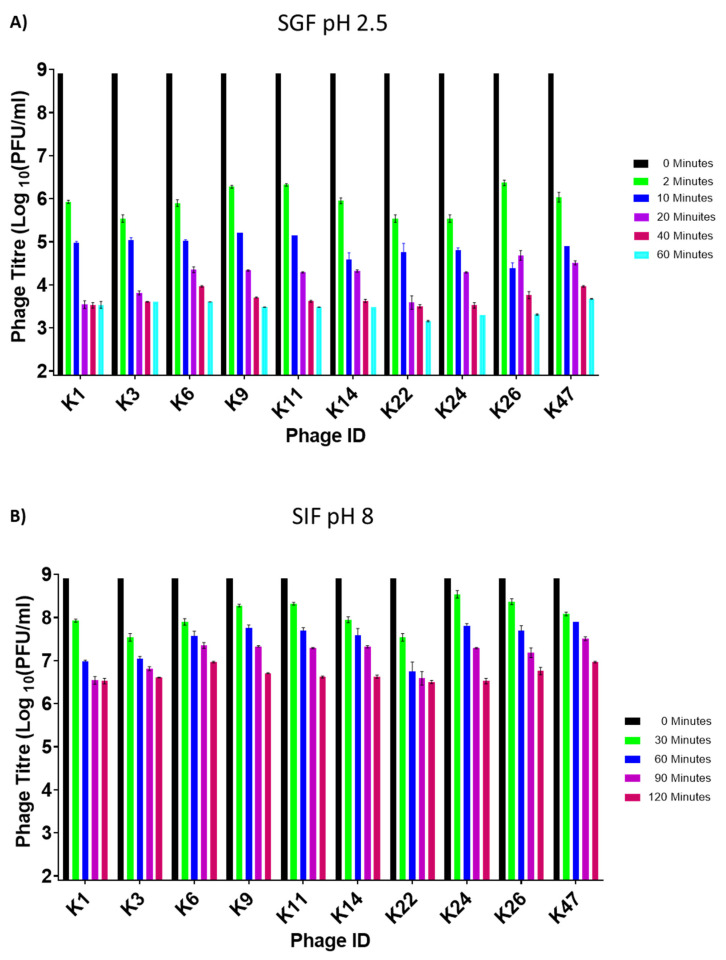
Phage stability in simulated gastric fluid (SGF) and simulated intestinal fluid (SIF). (**A**) Individual phage stability in SGF for 60 min of incubation at 42 °C. Phage titres for individual phages were determined after 0 (black bar), 2 (green), 10 (blue), 20 (purple), 40 (magenta), and 60 (turquoise) minutes. (**B**) Individual phage stability in SIF for 120 min of incubation at 42 °C. Phage titres for individual phages were measure after 0 (black bar), 30 (green), 60 (blue), 90 (purple), and 120 (magenta) minutes. All experiments were repeated twice and measured in triplicate. Each bar indicates phage titres at specific times. Error bars represent the standard error of the mean (±SE).

**Figure 4 viruses-14-01788-f004:**
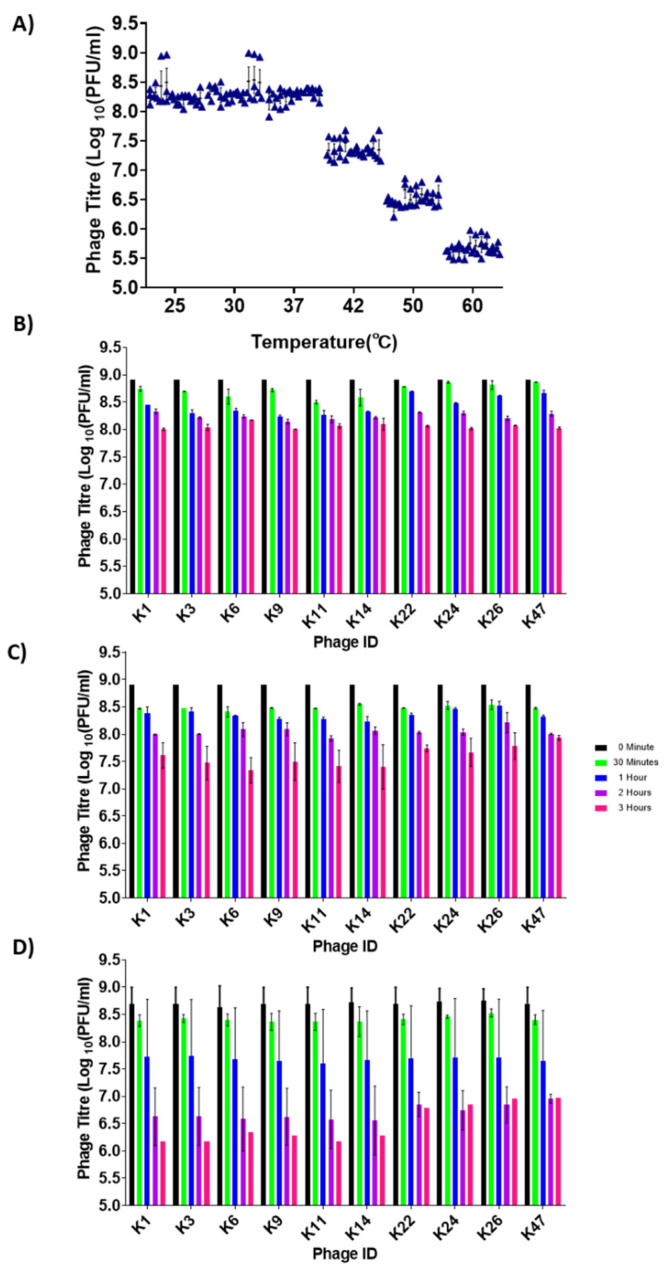
Phage thermal stability assay. (**A**) Overall stability of *S*. Enteritidis phages at 25 °C, 30 °C, 37 °C, 42 °C, 50 °C, and 60 °C for 12 h. The black triangle indicates individual replicate values of phages on each bar graph. The graph indicates phage titres after 12 h of incubation. (**B**) Phage stability at 37 °C after 3 h of incubation. The individual phage titres were determined after 0, 0.5, 1, 2, and 3 h. (**C**) Phage stability at 42 °C after 3 h incubation. The individual phage titres were determined after 0, 0.5, 1, 2, and 3 h. (**D**) Phage stability at 50 °C after 3 h of incubation. The individual phage titres were determined after 0, 0.5, 1, 2, and 3 h. All experiments were repeated twice and measured in triplicate. Error bars represent the standard error of the mean (±SE).

**Figure 5 viruses-14-01788-f005:**
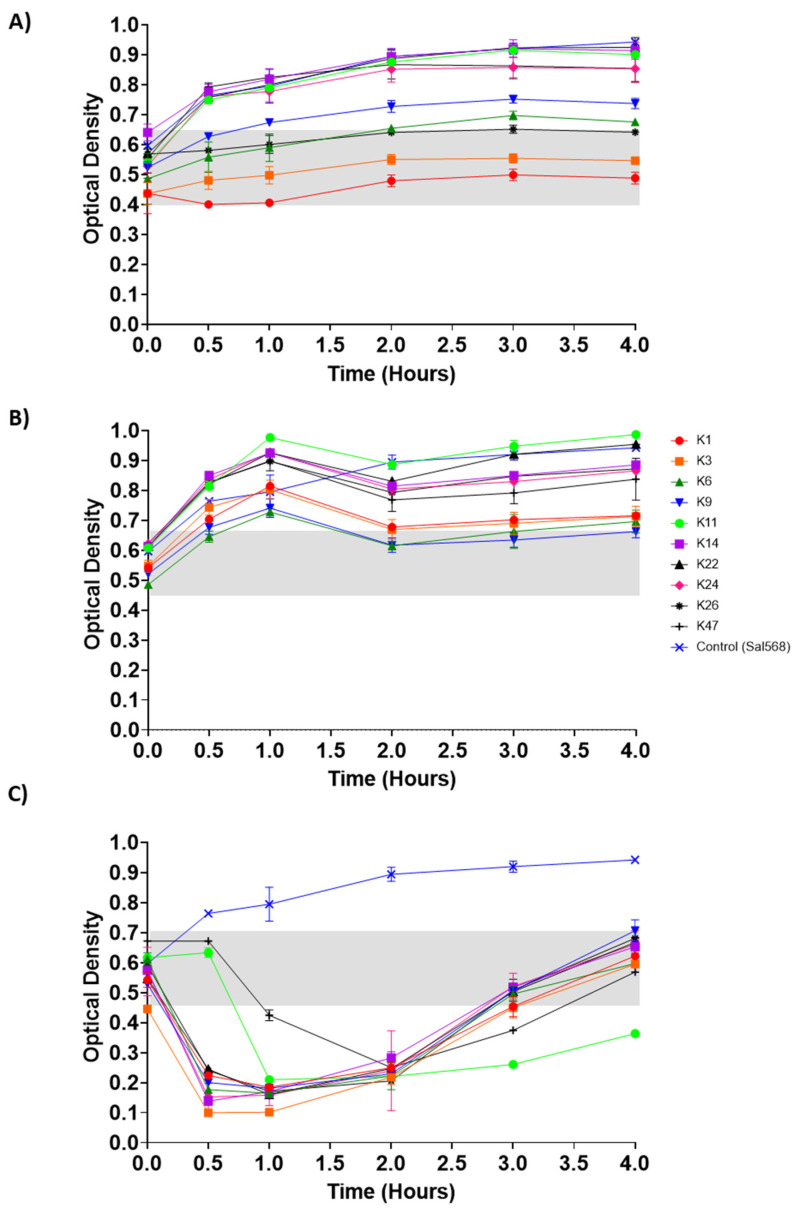
Control of *S*. Enteritidis growth by phages in pH-adjusted media. TSB was adjusted to pH (**A**) 2, (**B**) 3, and (**C**) 8. The optical density (OD_600 nm_) of the mixture of *S*. Enteritidis isolate 568 (10^6^ CFU/mL) and the 10 phages (4.5 × 10^7^ PFU/mL) were measured after 0, 0.5, 1, 1.5, 2, 2.5, 3, 3.5, and 4 h. The grey shading indicates initial OD values at the start of the experiment. All experiments were repeated twice and measured in triplicate. Error bars represent the standard error of the mean (±SE).

**Figure 6 viruses-14-01788-f006:**
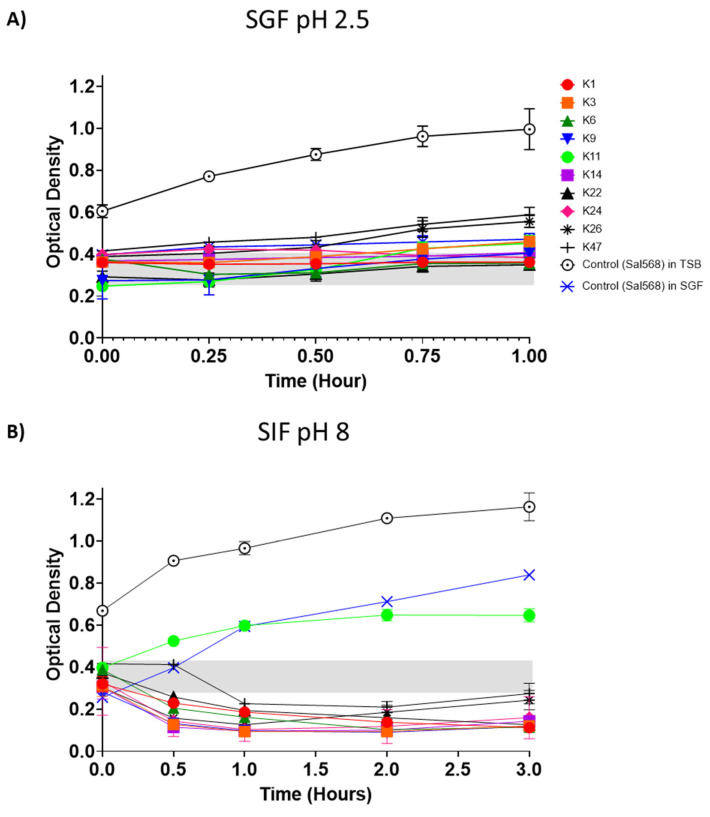
Control of *S*. Enteritidis growth by phages in SGF and SIF. (**A**) Effect of SGF on phage efficiency to control the growth of *S*. Enteritidis Sal568. (**B**) Effect of SIF on phage efficiency to control the growth of *S*. Enteritidis Sal568. The optical density (OD_600 nm_) of the mixture of *S*. Enteritidis Sal568 (10^6^ CFU/mL) and the 10 phages (4.5 × 10^7^ PFU/mL) were measured at 0, 0.5, 1, 1.5, 2, 2.5, 3, 3.5, and 4 h. The grey shading indicates initial OD values at the start of the experiment. All experiments were repeated twice and measured in triplicate. Error bars represent the standard error of the mean (±SE).

**Figure 7 viruses-14-01788-f007:**
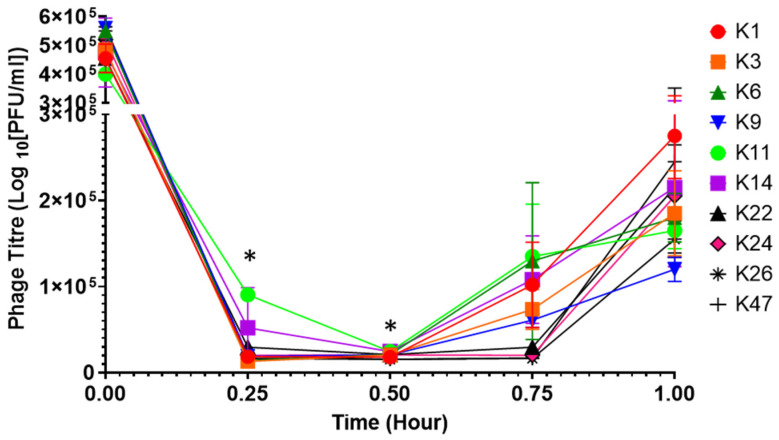
Phage replication in SGF. Phage titres measured following infection of *S*. Enteritidis Sal568 in SGF. Viral titre was determined through spot assays following the infection of *S*. Enteritidis isolate 568 (10^6^ CFU/mL) by the 10 phages (4.5 × 10^7^ PFU/mL). Error bars represent the standard error of the mean (±SE). All experiments were repeated twice and measured in triplicate. * Significant differences between phages ILRI_K1 and ILRI_K11 (*p* = 0.042), and between ILRI_K9 and ILRI_K11 (*p* = 0.0471) at 15 min; as well as between ILRI_K11 and ILRI_K26 (*p* = 0.0356) and between phages ILRI_K14 and ILRI_K26 (*p* = 0.0356) at 30 min of incubation.

**Figure 8 viruses-14-01788-f008:**
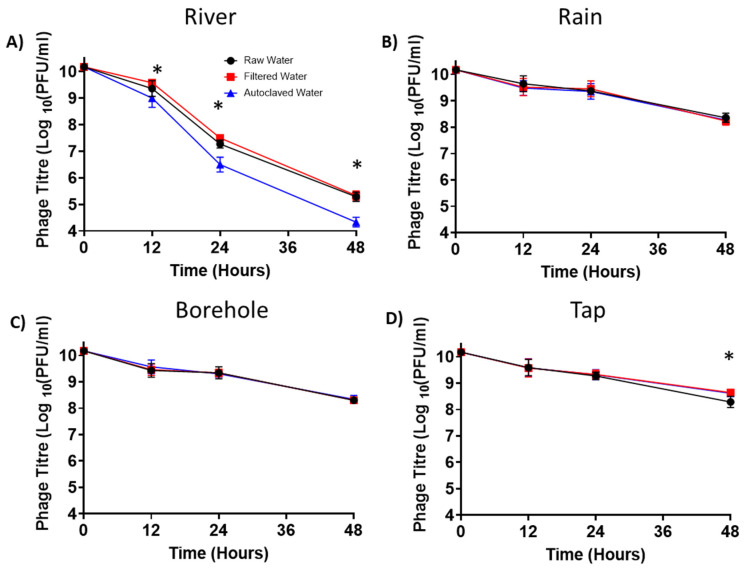
Phage persistence in water from different sources. Phage persistence in (**A**) river water and (**B**) rainwater. * Significant difference between filtered and autoclaved water, from 12 to 40 h of incubation (*p* values ranging from <0.0001 to 0.0396). Phage persistence in (**C**) borehole water and (**D**) tap water. * Significant differences between raw and filtered water at 48 h of incubation (*p* = 0.0365). Black circle: raw water, red square: filtered water, blue triangle: autoclaved water. All experiments were repeated twice and measured in triplicate. Error bars represent the standard error of the mean (±SE).

**Figure 9 viruses-14-01788-f009:**
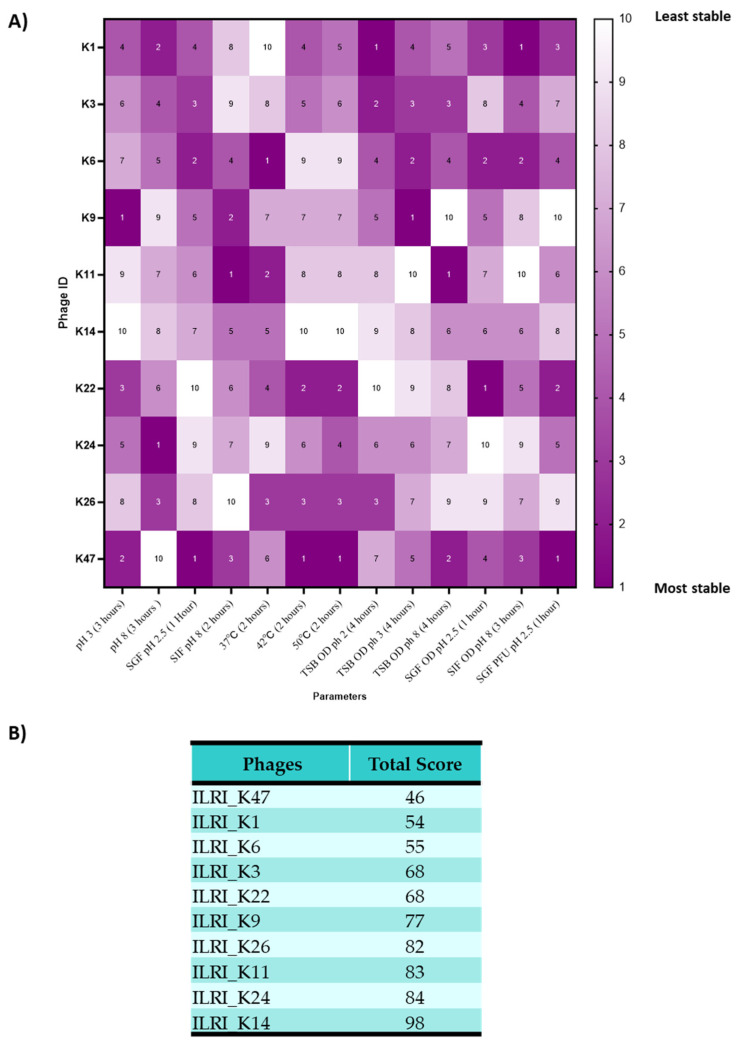
Phage stability scoring system. (**A**) The heatmap showing the ranking of the 10 *S*. Enteritidis phages based on the stability of 13 parameters, excluding water. Only the values at the end of the experiment were used for all these parameters. Deep purple indicates “most stable”, while white indicates “less stable”. All phages were ranked from the most stable (value of 1) to the least stable (value of 10) within a given parameter. (**B**) A table showing the ranking and scoring system for the phages. The total stability score was computed by adding individual scores for each parameter. The lowest value indicates the most stable phage.

**Table 1 viruses-14-01788-t001:** Typing of the *Salmonella* strains used in this study.

	Sal 16	Sal 73	Sal 157	Sal 172	Sal 177	Sal 181	Sal 182	Sal 187	Sal 188	Sal 192	Sal 194	Sal 312	Sal 568	Sal 569	Sal 571	Sal 572
*invA*	+	+	+	+	+	+	+	+	+	+	+	+	+	+	+	+
Group O (A-S)	+	+	+	+	+	+	+	+	+	+	+	+	+	+	+	+
Poly H (Phase 1 & 2)	+	+	+	+	+	+	+	+	+	+	+	+	+	+	+	+
Group D (9)	+	+	+	+	+	+	+	+	+	+	+	+	+	+	+	+
CRISPR 1	E	E	C	K	E	K	K	H	H	C	C	E	E	E	K	E
CRISPR 2	E	E	H	K	E	K	K	H	H	H	H	E	E	E	K	E

E: Enteritidis, C: Crossness, H: Heidelberg, K: Kentucky.

**Table 2 viruses-14-01788-t002:** Phages and *S*. Enteritidis host strains used in the phage isolation process.

Phages	Original *Salmonella* Strain	Identity of Poultry Farm (PF)/Slaughter House (SH)	Region	Remarks
ILRI_K1	Sal16	PF_16	Kiambu (Peri-Urban)	*Salmonella* isolated
ILRI_K3	Sal16	SH_6	Nairobi (Urban)	*Salmonella* absent
ILRI_K6	Sal16	SH_7	Nairobi (Urban)	*Salmonella* absent
ILRI_K9	Sal16	PF-58	Nairobi (Urban)	*Salmonella* absent
ILRI_K11	Sal73	SH_1	Kiambu (Peri-Urban)	*Salmonella* isolated
ILRI_K14	Sal73	PF_33	Kiambu (Peri-Urban)	*Salmonella* isolated
ILRI_K22	Sal177	PF_16	Kiambu (Peri-Urban)	*Salmonella* isolated
ILRI_K24	Sal177	SH_6	Nairobi (Urban)	*Salmonella* absent
ILRI_K26	Sal177	SH_6	Nairobi (Urban)	*Salmonella* absent
ILRI_K47	Sal312	SH_6	Nairobi (Urban)	*Salmonella* absent

**Table 3 viruses-14-01788-t003:** Nucleotide similarities between Kenyan phage genomes and published phage genomes from the NCBI public database.

	Most Similar Kenyan Phage Genome		Most Similar Phage Genome from the NCBI Public Database		
	Phage Name	Nucleotide Similarity % (Aligned Nucleotide %)	Phage Name	NCBI Accession Number	Nucleotide Similarity % (Aligned Nucleotide %)
ILRI_K1 *	ILRI_K22	99.99 (100%)	*Salmonella* phage wast	MT074451.1	93.72% (90%)
ILRI_K3	ILRI_K24	99.79% (100%)	*Salmonella* phage wast	MT074451.1	92.18% (90%)
ILRI_K6	ILRI_K24	98.23% (97%)	*Salmonella* phage wast	MT074451.1	92.30% (86%)
ILRI_K9	ILRI_K26	99.99 (100%)	*Salmonella* phage wast	MT074451.1	92.70% (90%)
ILRI_K11	None	--	*Salmonella* phage SP6	AY288927.2	89.67% (90%)
ILRI_K14	ILRI_K1 & _K22	97.27% (96%)	*Salmonella* phage wast	MT074451.1	93.59% (92%)
ILRI_K22	ILRI_K1	99.99 (100%)	*Salmonella* phage wast	MT074451.1	93.72% (90%)
ILRI_K24	ILRI_K3	99.79% (100%)	*Salmonella* phage wast	MT074451.1	92.41% (90%)
ILRI_K26 **	ILRI_K9	99.99 (93%)	*Salmonella* phage wast	MT074451.1	92.55% (90%)
ILRI_K47	ILRI_K9	96.74% (93%)	*Salmonella* phage wast	MT074451.1	92.44% (91%)

* One SNP variation in the genome of phage ILRI_K1 in comparison to the genome of phage ILRI_K22. ** The genome sequence of phage ILRI_K26 is incomplete and in 2 contigs. Both contigs aligned with the ILRI_K9 phage genome and one contig contain a SNP variation in comparison to ILRI_K9.

## Data Availability

Not applicable.

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
