# Peer review of "Salmonella Enteritidis Bacteriophages Isolated from Kenyan Poultry Farms Demonstrate Time-Dependent Stability in Environments Mimicking the Chicken Gastrointestinal Tract"

_viruses, 2022, doi:10.3390/v14081788_

Round 1

Reviewer 1 Report

Mhone et al. reports the isolation, identification and characterization of 13 phages that target Multi-drug resistant (MDR) Salmonella enterica Enteritidis, a pathogen of poultry farms. The pH and temperature stability of the isolated phages were extensively tested. Overall the research is still quite preliminary. The phage genomes should be sequenced and the genes encoded should be annotated.  Bacteria usually quickly develop resistance to phages, which hinder the therapeutical application of phages.  The combination of phages may be a solution to the resistance problem. They could test whether the combination of different phages is more efficient in killing the host. The therapeutical effects of the phages should be tested by using a animal model.

Author Response

We appreciate the reviewer’s comments as well as the grading of our papers (6 yes for the 6 questions). We certainly agree that these data do not represent the final story about these recently isolated phages and our ongoing project. However, what the reviewer is asking to provide is already in progress and will be presented in follow-up manuscripts which are at different stages of preparation. To summarize the ongoing work, the genomes of our 60+ phages have been sequenced already, but a substantial amount of bioinformatic analyses is ongoing to properly analyze the data and would be beyond the scope of this manuscript. Still, we have added additional info (where they were isolated) on the selected phages in Table 2 as well as better highlighting them in Figure 1.

We also agree with the reviewer that the best format for a phage-based product will be a cocktail. We are precisely working towards selecting the best phages to design an optimal cocktail for our strains of Salmonella and have identified candidates from our set of phages. However, selecting an efficient cocktail is a significant endeavor, which also includes their production in a suitable format. Again, this is a work in progress and shall be the subject of a subsequent manuscript.

 Finally, as mentioned by the reviewer, the in vivo work in an animal model is a key point. But again, this is a work in progress and cocktails need to be finalized and produced before being administered. We have just finished optimizing our Salmonella colonization model in chickens. We have started testing the persistence of our phages in vivo (without the bacterial host), and the paperwork has been submitted to our institutional animal care and use committee (IACUC) to test our phages for their capacity to decolonize Salmonella-inoculated chickens.

In our opinion, to have all the above new data in only one paper would be excessive. We want to highlight that this study’s objective was to look at the stability and persistence of phages in several environments relevant to chicken or poultry farming, first in vitro. Our manuscript already has 9 Figures (most of them are composite figures) and 2 tables (as well as Supplementary Figures). Therefore, we respectfully believe this is more than enough for a stand-alone paper. We hope the reviewer will understand that what is requested is simply not possible given the amount of work needed and the requested short time (20 days) to resubmit. 

Reviewer 2 Report

1.    Whether were phages isolated from the same chicken farm as Salmonella Enteritidis did? Authors should indicate it.

2.    Authors isolated several Salmonella Enteritidis stains from fecal matter, but why did choose salmonella strain sal568 to be infected?

3.    Authors tested the effect of pH, SGF, and SIF on phages’ capacity to control salmonella sp. in different circumstances. However, different environments would influence the growth of salmonella, thereby influencing the OD600. Therefore, how do authors figure out the impacts on growth of salmonella affected by settings or salmonella sp ?

Author Response

Thank you for your valuable comments. For point number 1, we mentioned that these phages were isolated from the same chicken farms as the Salmonella strains (formerly on line 222, now on line 225). However, we agree that it should be more clearly mentioned in the Materials and Methods. Therefore, we have added, in the section describing the isolation of phages (lines 115-116): “Phage samples were first obtained by inoculating feces, originating from the same chicken farms from which the Salmonella strains were isolated, into Tryptic Soy Broth (TSB).”

We must specify that about 66% of the farms had Salmonella-phages, but we were able to isolate Salmonella from only 10% of them. Therefore, we have also clarified in the main text: “Interestingly, we isolated Salmonella phages from about two-thirds of the visited farms while we could recover Salmonella sp. strains from only 10% of them were. [lines 222-224].

Concerning point #2, we had indicated in the text that we used Sal568 because the selected phages could infect this bacterial strain: “We used S. Enteritidis isolate 568 (Sal568) as it is sensitive to the selected phages.” [formerly line 325, now line 326]. We have now added in the Materials and Methods section under Bacterial strains: “The Sal568 strain was used to determine phage titers in subsequent experiments as it is sensitive to the selected phages.” [lines 109-111].

Finally, regarding point #3, the reviewer specifically refers to Figures 5 and 6. These experiments already contain an uninfected Salmonella control to assess the impact of these environments on bacterial growth. In the other figures, we were only interested to know how these environments solely affect the virions, irrespective of the presence of the host.

As for the English that could be improved as suggested by the reviewer, we want to specify that we used the Premium version of Grammarly, and the manuscript was also reviewed by professional English-speaking science editors and writers (Crayon Bleu, Inc., https://www.crayon-bleu.com/) before submitting the final version to the journal.

Round 2

Reviewer 1 Report

At least, the genome sequence information should be provided to prove that the phages reported here are unique isolates but not previously reported. 

Author Response

We have now provided a third table with the genome sequence information of our phages. The table describes the percent homology of the Kenyan phages between them as well as with known phages that have a genome sequence on NCBI. Our data indicate that these phages are unique isolates (with nucleotide similarities ranging from 89.67% to 93.72% with phage genomes from the NCBI public database) and have not been reported previously. However, the genome sequence information is not yet available for three of our 13 phages (ILRI_K10, ILRI_K23, and ILRI_K29). Therefore, since we cannot know at the moment if they are unique phage isolates, we have decided to remove the data regarding these three phages from all figures. We have also adjusted the main text where these phages were described and reported only on the 10 remaining phages for which we had genome sequence information. As explained in our previous reply to the reviewer, publishing the genomes of these phages will still require a substantial amount of time and will be the object of a subsequent publication. Regarding the English that could be improved as suggested by the reviewer, we want to specify that we used the Premium version of Grammarly, and the manuscript was also reviewed by professional English-speaking science editors and writers (Crayon Bleu, Inc., https://www.crayon-bleu.com/) before submitting the final version to the journal.